# Active Optimization of Chilled Water Pump Running Number: Engineering Practice Validation

**Shunian Qiu** [1] , **Zhenhai Li** [2], **Delong Wang** [3], **Zhengwei Li** [2] **and Yinying Tao** [4,*]

1 School of Civil Engineering and Architecture, Zhejiang University of Science and Technology, Hangzhou 310023, China
2 School of Mechanical Engineering, Tongji University, Shanghai 200092, China
3 Shanghai Discovery Energy Services Co., Ltd., Shanghai 201108, China
4 School of Design and Fashion, Zhejiang University of Science and Technology, Hangzhou 310023, China
* Correspondence: tao@zust.edu.cn

**Abstract:** To realize building energy conservation, appropriate operation of building energy systems is necessary. A chilled water pump, an essential component for chilled water transportation in building cooling systems, consumes substantial energy. Hence, its operation should be optimized. Previous studies on optimal pump control mostly focused on pump speed/frequency control, while the control of pump running number is usually too passive to realize energy-saving objectives. Moreover, existing relevant studies have some disadvantages, such as (1) too complex a workflow for maintenance; (2) dependence on accurate system performance models that take substantial data and labor to establish; and (3) high requirements on monitoring and sensors. To tackle those problems, this article proposes a simple, feasible approach to optimize the running number (on/off status) of chilled water pumps for building energy conservation. The proposed method is merely based on similarity/affinity laws and pump performance curves feasible for engineering practices. It has been implemented on a real cooling system in a battery factory. Our results suggest that: (1) based on similarity/affinity laws and pump performance curves, the estimation of potential targeted pump working points is accurate enough for optimal control (the flow rate estimation error is less than 2%, the frequency estimation error is less than 1 Hz); (2) the energy-saving effect of this control method is evident (20% of pump energy is saved by the proposed method compared to the former control logic); (3) the water grid operation condition is maintained well: cooling supply is not sacrificed by the control intervention (compared to the working condition before the intervention, grid pressure difference changed by 1.4%, flow rate changed by 2.6%). Regarding the low preconditions, simple workflow, and acceptable energy-saving performance of the proposed method, it is suitable for energy conservation in building cooling systems.

**Keywords:** chilled water pump; building cooling system; energy conservation; control practice

## 1. Introduction

### 1.1. Background

Building energy consumption is an important sector of society's total energy consumption [1,2]. To reduce building energy consumption, appropriate operation of building energy systems, especially cooling systems, is necessary. As an important component of the building cooling system, a chilled water pump (CHWP) makes chilled water flow in a chilled water loop, which realizes heat transfer from the user side to the source side. CHWPs consume substantial energy in building energy systems [3,4]. Hence, their optimal control/operation has been investigated by many researchers [5–8].

Typically, the operation of pumps can be adjusted via two controllable variables: pump on/off status, and pump working frequency [6]. In engineering practices, pump working frequency is usually adjusted by local PID (proportional–integral–differential) controllers

to maintain the chilled water grid pressure/pressure difference at a preset point [8]. Pump running number (i.e., on/off status of all installed pumps) is usually set in accordance with the chiller running number, or passively adjusted to avoid pump frequency exceeding the safe range.

### 1.2. Literature Review

Based on the conventional engineering control logic above, many researchers have carried out studies to further optimize the pump operation in building energy systems. Gao et al. [8] proposed a cascade control scheme to adjust the differential pressure set point at chilled water header pipes, based on which secondary CHWP frequency could be determined. Simulation case study results show that their proposed method could not only eliminate harmful deficit flow in primary–secondary chilled water loop but also save 26% of secondary CHWP energy consumption on a typical sunny summer day. It is worth noting that their proposed method does not include active optimization of pump running numbers: in their study, the pump running number is passively modulated in the abovementioned conventional way.

Ma et al. [9] improved the cascade control scheme in [8] by proposing a rule-based control scheme to regulate the temperature and flow rate in the chilled water grid. A one-year experimental case study indicated that their proposed control scheme could save over 30% of pump energy compared to conventional control logic. Similarly, their study focused on the adjustment of pump frequency instead of pump running number.

Sometimes, the operation of CHWPs is optimized along with other appliances in cooling systems. Shi et al. [10] adopted collaborated performance maps of different appliances (chillers, cooling towers, CHWPs, condenser water pumps) to simultaneously optimize the operation of a whole chilled water system. A simulation case study was conducted on the TRNSYS platform, and results suggested that the map-based control method could outperform the model-based control method and conventional control method (baseline) in energy conservation.

Table 1 lists several related studies on the optimized control of pumps in building cooling systems. As per Table 1, [6] presented and compared several examples of conventional control logic common in engineering practices, but not energy-efficient (only 4.5–10% of rated pump power could be saved); [8,11] proposed complex pump control methods; and [12,13] belong to the model predictive control (MPC) domain: they depend on accurate system performance models to function.

**Table 1.** Selected relevant studies on optimal pump control in buildings.

| Ref. | Online Real-Time Variable Monitoring | Basic Control Thinking | Control Signal | Energy Saving |
|---|---|---|---|---|
| [6] | Chilled water flow rate (header pipe) Grid pressure (header pipes) | **Passively** modulate the speed/frequency of single variable speed/frequency pump to meet system flow rate demand under moderate system changes. **Passively** control on/off status of pumps when the speed/frequency control above is not enough. | Pump frequency and on/off status | 4.5–10% of rated CHWP power |
| [8] | Chilled water flow rate (bypass pipe) Grid pressure (supply side and user side) Opening of user side AHU valves Supply air temperature on user side | Based on the monitored supply air temperature and valve opening, reset the set point of user side differential pressure. Based on the monitored bypass pipe flow rate, user side differential pressure, and its set point, reset the set point of pump pressure difference. Modulate pump speed/frequency with local PID controller to meet the pump pressure difference set point. **Passively** adjust pump running number only when pump speed/frequency would exceed the safe range. | Pump frequency and on/off status | 26% of CHWP energy on a typical summer day, compared to fixed differential pressure controller |

**Table 1.** *Cont.*

| Ref. | Online Real-Time Variable Monitoring | Basic Control Thinking | Control Signal | Energy Saving |
|---|---|---|---|---|
| [12] | Outdoor air dry bulb temperature<br>Outdoor air relative humidity<br>Diffuse horizontal radiation<br>Direct normal radiation | **Establish an artificial neural network (ANN) model based on historical system operation data** with outdoor temperature, outdoor humidity, radiation, and chilled water flow rate as input features, and room temperature as the estimated variable.<br>In real time control, predict room temperature with monitored environmental variables and controllable chilled water flow rate.<br>Along with the ANN, use genetic algorithm to search for the optimal chilled water flow rate to make the predicted room temperature meet its set point.<br>Modulate pump speed/frequency to meet the optimal flow rate demand. | Pump frequency | 51% of CHWP energy compared to simple on/off control logic |
| [11] | Chilled water temperature (supply side and user side)<br>Chilled water flow rate (bypass pipe and header pipes) | Reset the set point of user side chilled water temperature based on the supply side temperature.<br>Calculate the expected chilled water flow rate (header pipe) based on the reset temperature set point.<br>Determine the desired pump speed/frequency with PID feedback controller.<br>Modify the desired pump speed/frequency according to the measured bypass flow rate. | Pump frequency | 39% of CHWP energy over the whole year, compared to fixed-differential-temperature control logic |
| [13] | Dry-bulb temperature<br>Relative humidity<br>Solar irradiance<br>Datetime | **Establish gray-box models for each appliance in the cooling system based on historical data and regression, then link them to one integrated model.**<br>In real time control, estimate system energy consumption via the integrated model, along with monitored environmental variables and controllable CHWP frequency.<br>Use reduced gradient Frank–Wolfe algorithm to optimize CHWP frequency to minimize the estimated system energy. | Pump frequency | Hard to tell because pump optimization was not tested independently. |

### 1.3. Targeted Problem and Research Motivation

In engineering practices and reviewed studies above, pump running number is usually adjusted passively as a supplement to pump speed/frequency control [6,8]. However, according to fluid dynamics, a targeted pump working point (required pump head and flow rate) could be achieved with various pump running numbers. Hence, the pump running number could be actively adjusted to realize further energy conservation with no sacrifice on the operation of the chilled water grid.

Moreover, Table 1 indicates that current pump optimal control methods suffer from one or more shortcomings: (1) complicated algorithms and control logic that are not user-friendly for control program development and maintenance [8,11]; (2) dependence on predefined system performance models (which take substantial data and labor to establish) for optimal control [12,13]; (3) high requirements on real-time system operation monitoring (i.e., additional variables are required to be monitored, which takes supplementary sensor installation) [8].

This research intended to tackle the existing problems above and further utilize the energy-saving potential of pump running number modulation. In other words, this article proposes a simple, user-friendly, engineering-feasible method to optimize the running number of CHWPs. Section 2 demonstrates the workflow of the proposed control method, Section 3 elaborates the engineering implementation of the proposed method on a real cooling system of a battery factory in Fujian province, China, Section 4 evaluates the performance of the proposed method based on both short-term and long-term system operational data after the control intervention, Section 5 concludes the paper with the discussion on the application prospect, engineering compatibility and future works of the proposed method.

## 2. Methodology

### 2.1. Overview

Before introducing the proposed method, here are some critical assumptions, limitations, and preconditions of this method.

(1) ***Offline preparation:*** The optimization process of the proposed approach is based on pump performance curves and similarity/affinity law, so the performance curves of the targeted pumps should be available before the application of the proposed method. Specifically, nominal/reference pump head curve (i.e., pump flow rate versus pump head) and nominal/reference pump power curve (i.e., pump flow rate versus pump input electrical power) are needed (they are usually provided in the equipment manual). Both curves are assumed to be accurate by default. In addition, the safe range of pump operating frequency is required, which is usually available in pump manuals.

(2) ***Compatible system type:*** This method is compatible with the differential pressure-based chilled water system, in which pump speed/frequency is controlled to maintain the pressure difference between chilled water header pipes at the set point [8]. If the targeted system is based on differential temperature to control pump speed/frequency (not so common) [13], then the proposed method cannot be applied without modification.

(3) ***Requirements on the real-time monitoring/measurement:*** The proposed method requires real-time data of header pipe chilled water flow rate, grid differential pressure, and the set point of grid differential pressure. These variables are typically monitored by default in real systems.

The basic thinking of the proposed method is to calculate the expected pump frequency and electrical power corresponding to each potential pump running number. After that, we could find the optimal pump running number with the minimal expected electrical power. The energy-saving potential of CHWPs could be explained by Figure 1:

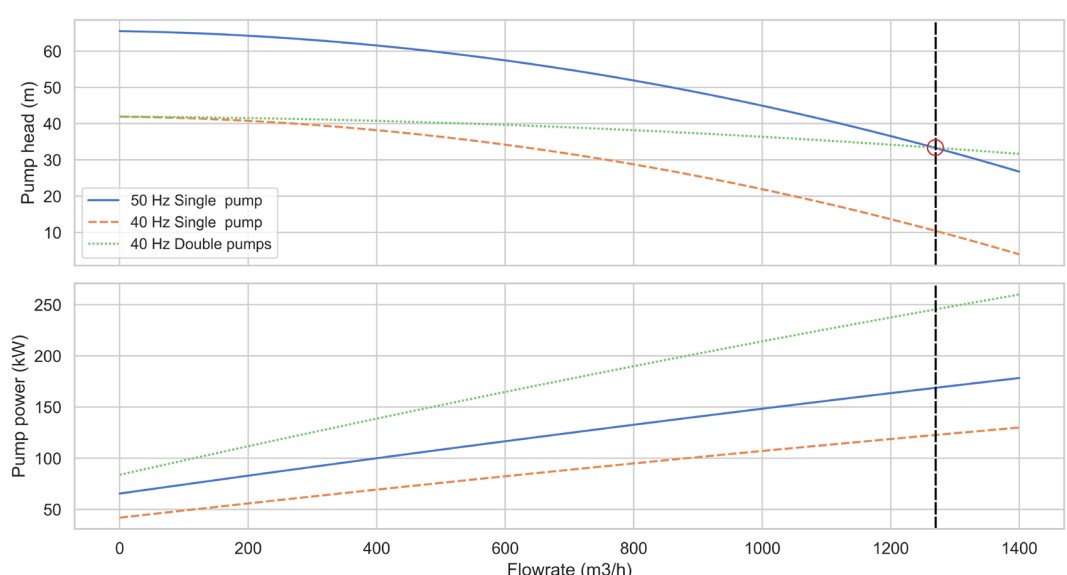

**Figure 1.** Pump operational curves under different scenarios.

Figure 1 is plotted according to the equipment manual of the case system in Section 3. Figure 1 shows the pump operational curves under three different scenarios: 50 Hz–one pump, 40 Hz–one pump, and 40 Hz–two pumps. To reach one working point (i.e., an expected pump head and flow rate, which is marked by a red circle), there could be multiple pump operating plans: fewer pumps with higher frequency or more pumps with lower frequency. Moreover, different operating plans lead to different electrical power consumptions, which implies the energy saving potential via optimal control. And the method proposed in this study is intended to tap this potential.

### 2.2. Calculation Process

As illustrated in Figure 2, the optimal pump running number is determined by the following steps.

First, for chilled water loops, usually chilled water pumps work in the region of quadratic resistance law (or approximately in this region) [14]. Hence, the grid resistance $S_{grid}$ (kg/m$^7$) could be deduced by Equation (1) with monitored header pipe flow rate $Q_{measure}$ (m$^3$/s) and the pressure difference on the header pipe $\Delta P_{grid}$ (Pa)

$$\Delta P_{grid} = S_{grid} Q_{measure}^2 \tag{1}$$

Then, since the proposed method is designed for differential pressure-based chilled water systems, there would be a set point of the grid pressure difference $\Delta P_{set,grid}$ (Pa) accessible in the on-site local control system. Hence, the needed header pipe flow rate $Q_{need}$ (m$^3$/s) could be determined with Equation (2) along with the $\Delta P_{set,grid}$ and the deduced $S_{grid}$.

$$\Delta P_{set,grid} = S_{grid} Q_{need}^2 \tag{2}$$

Next, traverse the pump running number $n$ from one to $N$ ($N$ is the total installed pump number), and **repeat the following calculation steps with each potential pump running number** $n$.

Equally distribute $Q_{need}$ to $n$ running pumps to get the single pump flow rate $Q_{single}$ (m$^3$/s) because parallel pumps equally undertake system's total flow rate [14].

Considering the pressure drop between pressure meter and pump inlet/outlet (mainly caused by the filter at the pump inlet), the targeted grid pressure difference $\Delta P_{set,grid}$ differs from the targeted pump head ($\Delta P_{set,pump}$, Pa). Hence the targeted pump head should be calculated with Equation (3). Note, $S_{filter}$ (kg/m$^7$) is the resistance of the pump inlet filter, which is regarded as a static parameter herein and should be determined with some simple tests in advance. Details are given in Section 3.

$$\Delta P_{set,pump} = \Delta P_{set,grid} + S_{filter} Q_{single}^2 \tag{3}$$

When $S_{grid}$ and $S_{filter}$ stay constant, the pump's working points are similar under different frequencies (i.e., Equation (4c)) [14]. In addition, nominal/reference and actual working points are subject to Equations (4b) and (4a), respectively. Based on similarity/affinity law (Equation (4c)) and the known nominal/reference pump head curve (Equation (4b)), Equation (4d) could be derived. Then we could solve the targeted pump frequency $f_{set}$ (Hz) with Equation (4d), where $f_{ref}$ is the nominal/reference pump working frequency (Hz), $A_{ref}, B_{ref}, C_{ref}$ are pump head curve coefficients under the nominal/reference frequency [15]. Parameters with subscript "ref" could be acquired from the equipment manual. While $A_{set}, B_{set}, C_{set}$ are coefficients under the targeted working frequency, they are not required in this method. In brief, Equations (4a) to (4c) are for formula derivation, readers could use Equation (4d) directly

$$\begin{cases} \Delta P_{set,pump} = A_{set} Q_{single}^2 + B_{set} Q_{single} + C_{set} & \text{(4a)} \\[2mm] \Delta P_{ref,pump} = A_{ref} Q_{ref}^2 + B_{ref} Q_{ref} + C_{ref} & \text{(4b)} \\[2mm] \left(\dfrac{Q_{ref}}{Q_{single}}\right) = \left(\dfrac{f_{ref}}{f_{set}}\right) = \sqrt{\dfrac{\Delta P_{ref,pump}}{\Delta P_{set,pump}}} & \text{(4c)} \\[2mm] \Delta P_{set,pump} = A_{ref} Q_{single}^2 + B_{ref} Q_{single} \times \left(\dfrac{f_{set}}{f_{ref}}\right) + C_{ref} \times \left(\dfrac{f_{set}}{f_{ref}}\right)^2 & \text{(4d)} \end{cases}$$

Again, based on similarity/affinity law, estimate total pump power $W$ (kW) under the targeted working point using Equation (5), where $n$ is the potential pump running

number being evaluated currently, $D_{ref} \sim G_{ref}$ are pump power curve coefficients under the nominal/reference frequency. They could be acquired from the equipment manual.

$$W = n \times \left[ D_{ref}Q_{single}^3 + E_{ref}Q_{single}^2 \times \left( f_{set}/f_{ref} \right) + F_{ref}Q_{single} \times \left( f_{set}/f_{ref} \right)^2 + G_{ref} \times \left( f_{set}/f_{ref} \right)^3 \right] \tag{5}$$

Finally, after the calculation of targeted working points under all potential pump working numbers, find out the minimum targeted pump power $W$ within the safe frequency range (usually 35–50 Hz). The pump working number corresponding to this control plan is the optimal solution.

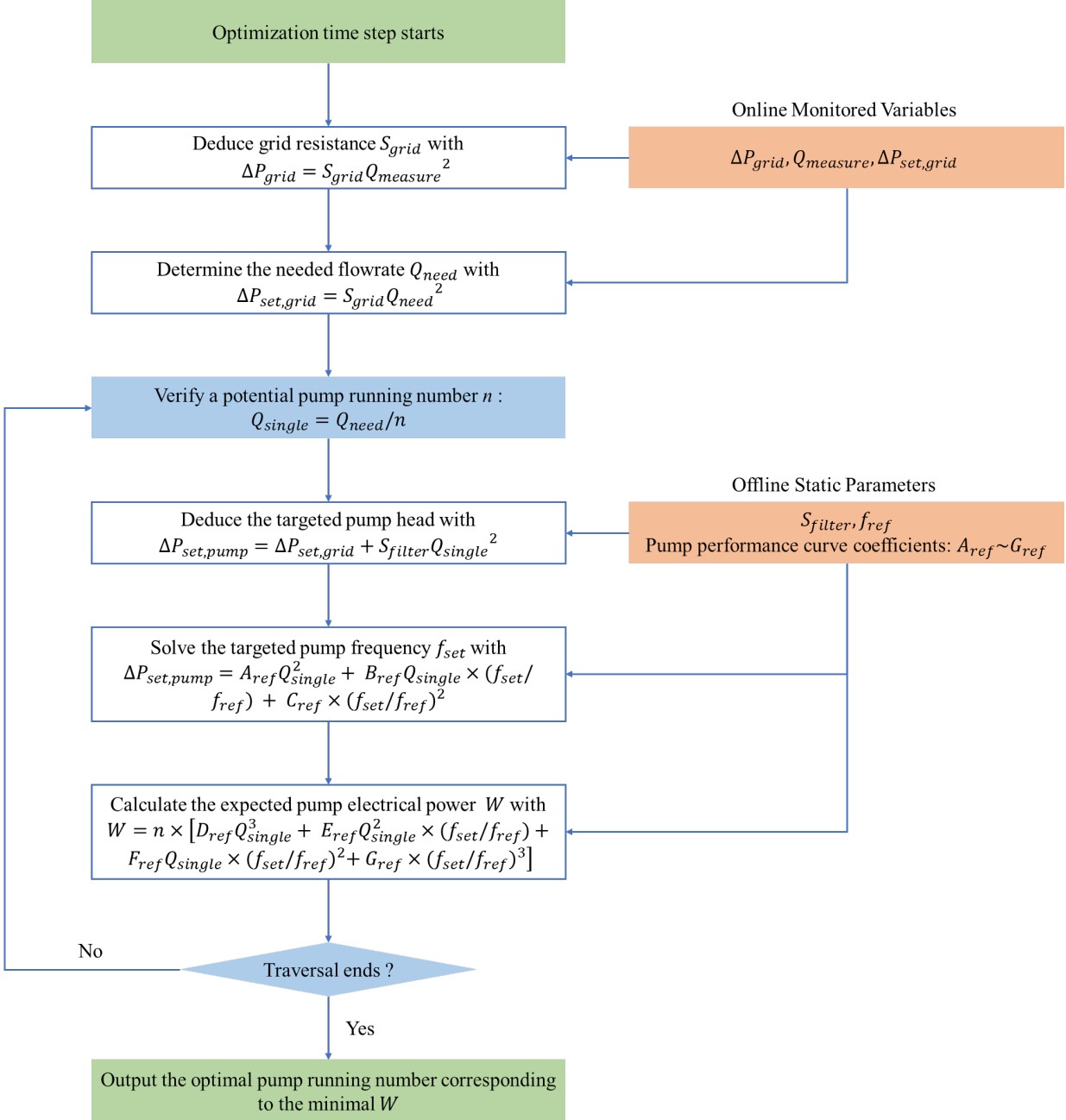

**Figure 2.** Workflow of the proposed optimal control method.

### 3. Engineering Practice Case Study

Although many studies choose to analyze chilled water systems via simulation for variable control [5,8], the case study in this article is conducted on a real cooling system of

a battery factory in Fujian province, China. We implemented the proposed pump control method on the case system to find out the in situ performance of the method. The case system is composed of six water-cooled chillers, six primary CHWPs directly linked to six chillers and ten identical secondary CHWPs for user-side chilled water transportation. The system layout is illustrated in Figure 3. Before our intervention, the case system controls secondary CHWP frequency by a local PID controller to maintain constant differential pressure on header pipes; only when speed/frequency control cannot meet the pressure requirement, the pump running number would be changed passively to compensate. The running number of secondary CHWPs is to be optimized in this case study, while pump working frequency would still be controlled by the local PID controller. Pumps' nominal/reference performance curves (pump head curve and power curve under 50 Hz) are illustrated in Figure 1. From the head curve in Figure 1, seven coefficients ($A_{ref}, B_{ref}, C_{ref}, D_{ref}, E_{ref}, F_{ref}, G_{ref}$) could be determined by curve regression. Usually, pump head curves provided by manufacturers are accurate enough for pump modeling [16].

Moreover, the parameter $S_{filter}$ needs to be determined with online tests: first, initialize $S_{filter} = 0$; then program the proposed method with the determined $A_{ref}, B_{ref}, C_{ref}$ and initial $S_{filter}$; run the code to estimate pump frequencies under each pump running number; check the estimated frequency value and the current measured value; tune the $S_{filter}$ value to make the estimation results match the measurement.

Figure 4 illustrates the data flow in this case study. As shown in Figure 4, in every control time step (1) the PLC (programmable logical controller) collects operational data from the pump variable frequency drive (VFD) and sensors (i.e., pressure meters, flow meters); (2) system operational data is transmitted to the cloud server through a gateway; (3) the data are archived in the InfluxDB database and then fetched by the core control program; (4) the control program determines a suggested pump running number (control signal) and sends it back to the gateway; (5) the control signal transferred by the gateway and the PLC would be executed on the pump VFD (Table 2).

**Table 2.** Secondary CHWP characteristics.

| Model | Flow Rate (m³/h) | Head (m) | Installed Number | Input Electrical Power (kW) | Efficiency (%) |
|---|---|---|---|---|---|
| KQSN300-M9/423 | 850 | 50 | 10 | 160 | 84.9 |

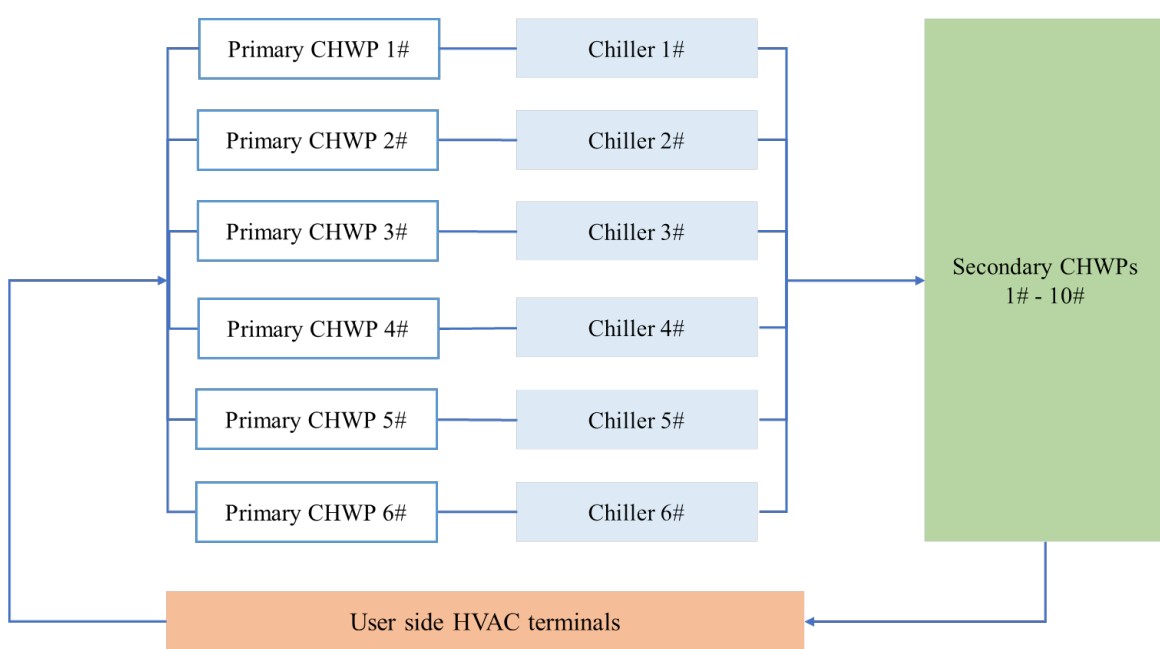

**Figure 3.** Case system layout.

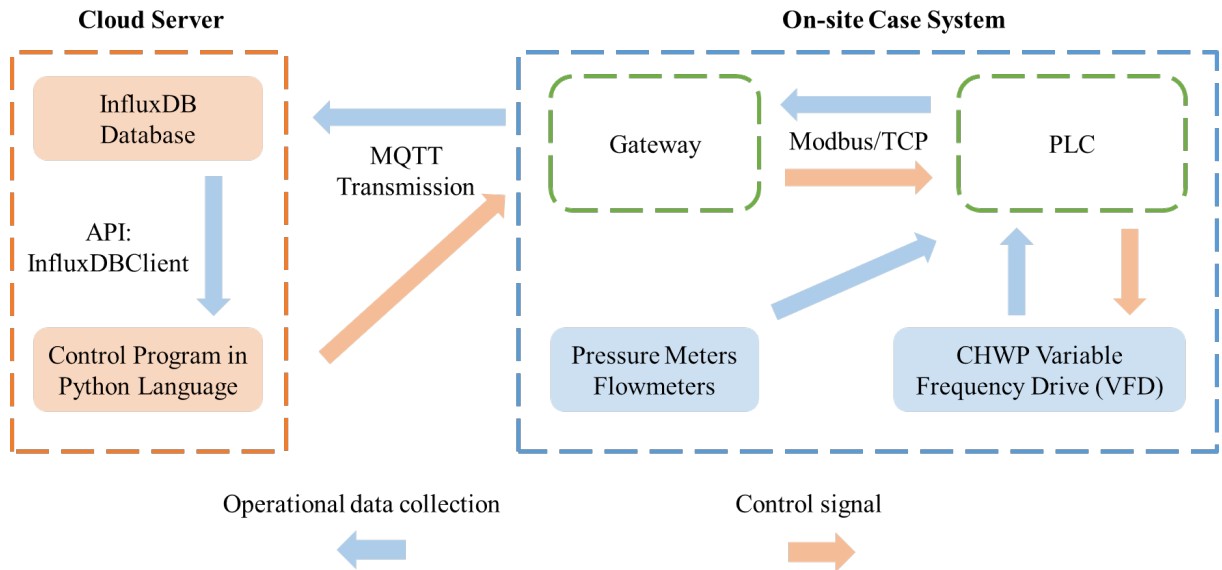

**Figure 4.** Data flow in the case study.

## 4. Results and Discussion

The application was realized on 17 December 2021. Before the application of the proposed method, the operation of secondary CHWPs in the case system was not optimized. These pumps typically worked at high frequencies with few running numbers. For instance, from 15 December to 17 December, frequency data of CHWPs mainly lay within 43–45 Hz (Figure 5), which is empirically not efficient for pump operation [11].

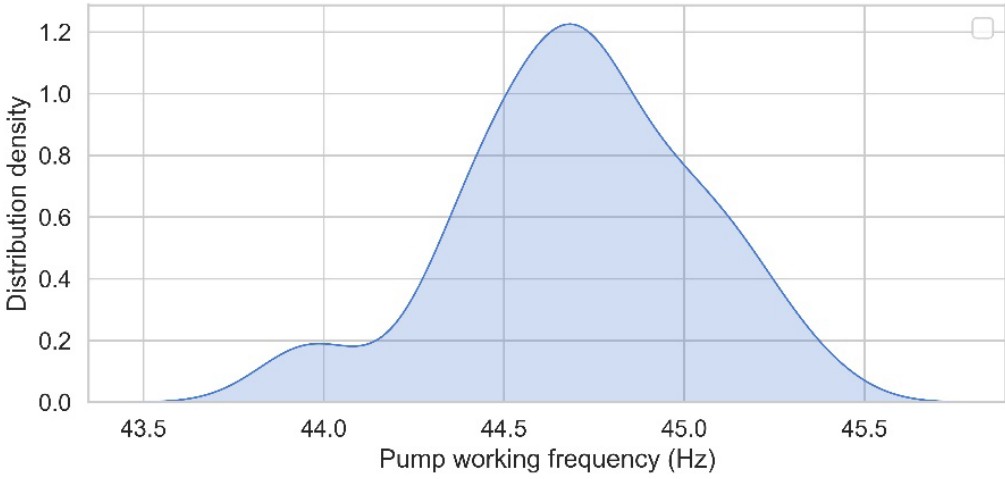

**Figure 5.** Pump working frequency distribution before optimal control intervention.

### 4.1. Short-Term Performance

During the optimization process, the proposed method estimated potential pump power and frequency values corresponding to all potential pump running numbers. The estimation results indicated that it would be better to increase the pump running number from five to six. After the intervention of the proposed method, the number of running pumps was quickly enhanced, and pump power decreased with frequency. Detailed system working conditions before and after the control action are addressed in Table 3.

**Table 3.** Real-time system conditions during the control practice.

| Monitored Variables | Before Intervention 14:55, 17th December | Optimization Result (Estimation) | After Intervention 14:59, 17th December |
|---|---|---|---|
| $\Delta P_{grid}$ (Pa) | 78,889 | 80,000 | 80,000 |
| $Q_{measure}$ (m$^3$/s) | 4636 | 4669 | 4758 |
| Secondary CHWP running number | 5 | 6 | 6 |
| Secondary CHWP frequency (Hz) | 44 | 38 | 39 |
| Total input power of secondary CHWPs (kW) | 601 | / | 476 |

Table 3 suggests that:

(1) By comparing the estimated values and the corresponding measured values (after control intervention), it could be seen that the flow rate estimation error is less than 2%, and the frequency estimation error is less than 1 Hz. Hence, the estimation accuracy of the proposed methodology (based on similarity/affinity and pump head curves) is acceptable for optimal control.

(2) About 20% of the pump energy is saved with the proposed method (601–476 kW), which validates the energy-saving performance. The energy reduction is not as significant as results in existing pump frequency control studies (over 30%) [9,11]. But since the targeted control signals are different (running number/operating frequency), it would be not proper to simply compare the energy-saving amount between this study and the other two. Moreover, our proposed methodology is simpler than those in [9,11], which suggests its better feasibility in engineering practices.

(3) Compared to the working condition before the intervention, grid pressure difference changed by 1.4% and flow rate changed by 2.6%, which shows that the water grid operation condition is maintained well; cooling supply is not sacrificed for energy conservation by the control intervention. Moreover, the stable operation of the whole chilled water system means the proposed method would not affect the efficiency of other relevant appliances. Compared to [8,9,11], whose approaches involve the adjustment of the grid flow rate and differential pressure, the method of this study may be more suitable for industrial constructions which requires higher operating stability.

### 4.2. Long-Term Performance

The optimal control program kept running on the case system since 17 December And the operation data from 17 December to 31 December is illustrated in Figure 6. As shown in Figure 6, after the intervention of the proposed method, the running number of secondary CHWPs is dynamically adjusted to meet the real-time demand ($\Delta P_{grid}$ and $Q_{measure}$) of the case system. In addition, the working frequencies of pumps are controlled to a range of 30–42 Hz, significantly lower than before. In doing so, the pump energy consumption was reduced compared to the preintervention period, which is validated in Figure 7. It can be seen in Figure 7 that before 17 December, the pump power of the case system barely varied with the dynamic system condition, which means that the old control logic was not reasonable enough to utilize the energy-saving potential of these CHWPs. On the contrary, after the control intervention, the proposed optimized control logic dynamically adjusted the CHWP running number, which led to lower energy consumption (the pump electrical power after 17 December is lower than before).

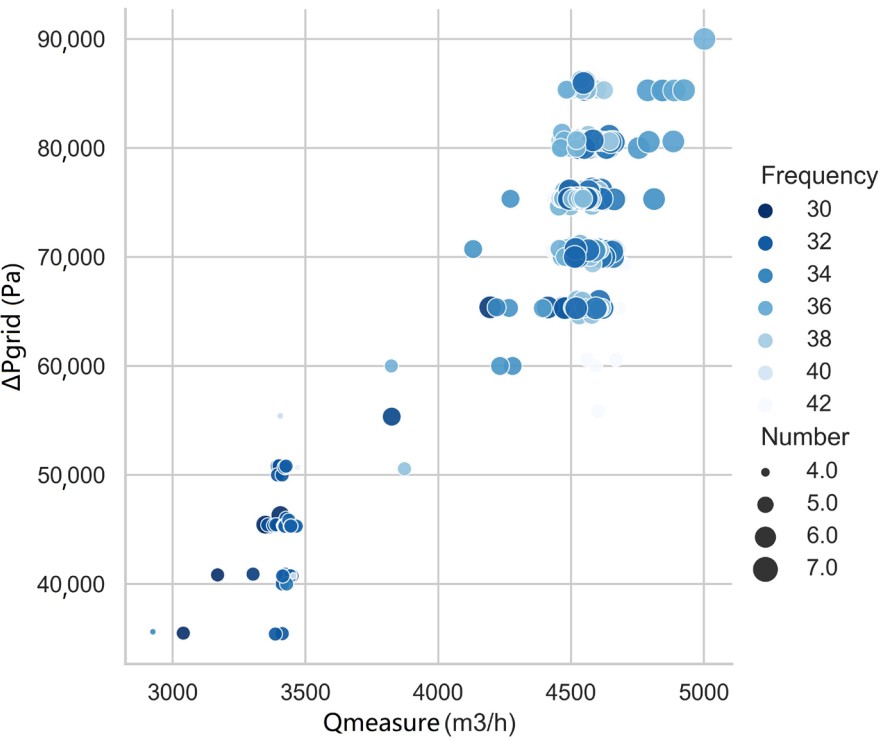

**Figure 6.** Pump operation after optimal control intervention.

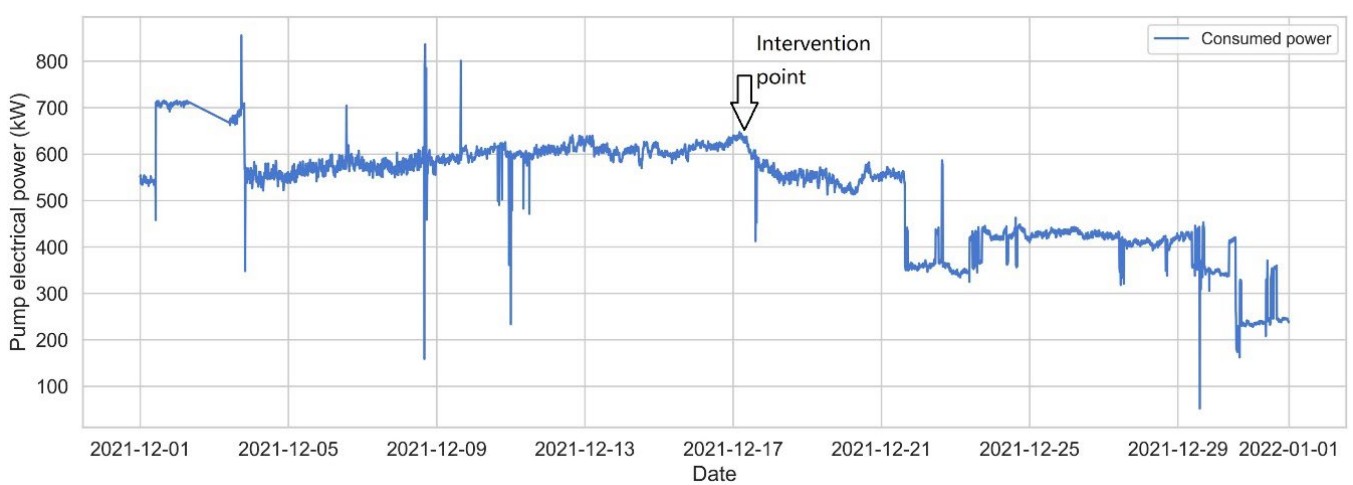

**Figure 7.** Time series of pump electrical power.

## 5. Conclusions and Future Work

### 5.1. Conclusions

*Research motivation:* Building energy conservation is important for the whole society. CHWPs consume substantial energy in building energy systems [5]. Optimization of their operation is valuable and necessary. Relevant studies mainly focus on the optimization of pump frequencies, while the active adjustment of pump running numbers is often neglected [6,11]. Moreover, they have some disadvantages, such as (1) too complex a workflow for maintenance; (2) dependence on accurate system performance models, which take substantial data and labor to establish; and (3) high requirements on monitoring and sensors. To further utilize the energy-saving potential of CHWPs in building energy systems, this article presents a simple control method to optimize the pump running number.

*Engineering performance:* Based on laws of similarity/affinity and pump performance curves, the proposed method has been applied to control a real chilled water system in a

battery factory. The application results suggest that this method could accurately estimate targeted pump working conditions under different running numbers, based on which the pump running number was optimized to reduce pump electrical power by 20%. The grid operation status was kept stable, hardly influenced by the control action. That proves the effectiveness of the approach in industrial constructions.

*Compatibility and feasibility:* Currently many relevant studies on optimal pump control tend to develop complex and complete control logic to maximize the energy saving performance [6,8,9,11–13], this kind of thinking brings (1) more preconditions for deployment (more variables need to be monitored, more offline information/models need to be ready in advance); (2) algorithm complexity means difficulty in maintenance. These disadvantages prevent many novel control methods being applied widely because engineering systems usually cannot provide sufficient conditions as laboratories do. Hence, the simple methodology and low preconditions of the method proposed herein renders its compatibility and feasibility in engineering practices. Moreover, the on-site performance proves that the proposed method could cooperate well with existing pump speed/frequency control logic of the case system.

*Application prospects:* It is worth noting that the proposed method has also been successfully applied to another cooling system in Jiangsu province, where its control performance is validated again (the implementation process of this Jiangsu project is recorded and uploaded to our website, please refer to the Appendix A), and when promoting these two projects (in Fujian and Jiangsu), we realized that the application of the proposed method is suitable to work with online building energy management platforms, which are popular these days:

(1)   the proposed method is deployed in the cloud server of the cooperating energy management platform (Figure 4);
(2)   by default, energy management platforms always need to build data collection-transmission frameworks on client buildings for energy monitoring and management;
(3)   real-time monitored variables required by the proposed method are also important and accessible for conventional pressure-based pump control and energy management platforms.

Hence, integrating the proposed method to current energy management platforms does not bring much marginal cost, except for commissioning the control program on the cloud server. In brief, the proposed method is suitable for existing energy management platforms as a cost-efficient component/module.

### 5.2. Future Works

As introduced in the beginning of the Section 2, the proposed method is designed for differential pressure-based chilled water systems where pump speed/frequency is controlled based on the measured pressure difference. Although currently this type of chilled water systems is the most common in engineering practices, there are more and more novel methods being invented by researchers to optimize pump speed/frequency [11,17]. Hence, it would be interesting to investigate how to integrate the proposed method with novel pump speed/frequency control methods. In doing so, the energy efficiency of pumps could be further enhanced.

As demonstrated in the Section 2, the proposed method requires accurate pump performance curves to function. Since pump performance would degrade with time, original pump performance curves could become biased [18]. Hence, it would be useful to embed an online self-calibration mechanism in the proposed method to improve its long-term control performance.

**Author Contributions:** Conceptualization, S.Q. and D.W.; methodology, S.Q.; software, S.Q. and D.W.; validation, S.Q. and D.W.; formal analysis, S.Q.; investigation, Z.L. (Zhengwei Li); resources, D.W.; data curation, D.W.; writing—original draft preparation, S.Q.; writing—review and editing, S.Q.; visualization, S.Q.; supervision, Y.T. and Z.L. (Zhenhai Li); project administration, Y.T. and Z.L. (Zhenhai Li). All authors have read and agreed to the published version of the manuscript.

**Funding:** This research received no external funding.

**Institutional Review Board Statement:** Not applicable.

**Informed Consent Statement:** Not applicable.

**Conflicts of Interest:** The authors declare no conflict of interest.

## Appendix A

Another on-site engineering practice of this method in Jiangsu Province is recorded as a video. Please check http://doi.org/10.13140/RG.2.2.17227.87840 for more information.

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
