# Peer review of "Active Optimization of Chilled Water Pump Running Number: Engineering Practice Validation"

_sustainability, doi:10.3390/su15010096_

Round 1

Reviewer 1 Report

The paper entitled” Active optimization of chilled water pump running number: Engineering practice validationdeals with a very interesting topic, and it included interesting ideas.

However, I have the following comments that hopefully help the authors improve their paper:

·       The research question must be better contextualized and be more convincing. What is research gap? How will this research fill the gap? The contribution of research must be more highlighted.

·       The structure (outline) of the paper could be given at the end of the introductory chapter.

·       The introduction is mixed with a short literature review. I suggest to the authors a section dedicated to literature review where should analyse the existing works in the way to show the gap in the literature compared to this work. Furthermore, it would be better if authors can have a table comparing the closely related works on various dimensions and clearly showing the contribution of the paper.

·       I suggest that the authors add a research method diagram. This will provide a snapshot of the research steps followed and will help the reader in a clearer understanding of the paper.

·       What are the limitations of the study in terms of the proposed method, data used, approaches, and/or analysis?

·       The authors should convince the readers, that their contribution is so important. These issues deserve a deeper discussion: What are the managerial implications from this work? What are the implications for theory and practice? How does this understanding help organizations and the industry to make better decisions? How decision or policy makers could benefit from this study.

·       The conclusions are slight. Furthermore, it could be interesting to discuss in the conclusion part, the future work, several potential futures research should be addressed.

·       As usual a final thorough proof-reading is recommended.

 I encourage the author to think along those questions and to develop this work further along those lines.

Author Response

Thank you for your prompt review, Please see the attachment..

Reviewer 2 Report

This manuscript presents a simple approach to optimize the running number (on-off statuses) of chilled water pumps for building energy conservation. The proposed method is merely based on similarity laws and pump head curves, feasible for engineering practices. The proposed method has been implemented on a real cooling system in a battery factory. This manuscript has good application value. However, this manuscript still has the following problems to be improved.

1. Firstly, the traditional methods and their performance should be explained in the introduction. Authors need to point out problems with existing research.

2. What are the scientific questions of this study? This needs to be made clear in the introduction.

3. Should the theoretical basis of the optimization method proposed in this study be revealed in detail in the paper?

4. Limitations of the study need to be described in detail.

Author Response

Thank you for your careful review, Please see the attachment.

Round 2

Reviewer 1 Report

The manuscript has significantly improved as compared to the previous version. Indeed, the authors tried to improve it, and the main weaknesses are solved. 

Thus, in my opinion, the manuscript is on the borderline recommendable for publication.